Effectiveness of telenutrition in a women’s weight loss program

Kuzmar Isaac E. 1 isaackuzmar@yahoo.es
Cortés-Castell Ernesto 2
Rizo Mercedes 1
1 Department of Nursing, Faculty of Health Sciences, University of Alicante , Alicante , Spain
2 Department of Pharmacology, Pediatrics and Organic Chemistry, School of Medicine, Miguel Hernández University , Alicante , Spain
Nock Nora
Electronic publication date: 2015 Feb 3
Publication date: 2015
Volume: 3
Electronic Location ID: e748
Received 2014 Nov 7; Accepted 2015 Jan 13
Copyright: © 2015 Kuzmar et al.
Copyright year: 2015
Copyright holder: Kuzmar et al.
License: This is an open access article distributed under the terms of the Creative Commons Attribution License, which permits unrestricted use, distribution, reproduction and adaptation in any medium and for any purpose provided that it is properly attributed. For attribution, the original author(s), title, publication source (PeerJ) and either DOI or URL of the article must be cited.
License URL: https://creativecommons.org/licenses/by/4.0/

Keywords: Telenutrition, e-health, Obesity, Overweight

Funding: The authors declare there was no funding for this work.

==============================
Objective. The objective of this study is to evaluate the effectiveness of telenutrition versus traditional nutritional consultations for female obese patients in need of nutritional treatment.

Methods. A comparative clinical study was conducted among 233 obese or overweight women (including 20 who dropped out and 60 who failed) who consulted a nutrition clinic in Barranquilla (Colombia) for nutritional assessment and chose either telenutrition or a traditional consultation that included a weekly follow-up consultation over 16 weeks, food consumption patterns, Body Mass Index (BMI, kg/m2) registeration and waist and hip circumference registeration. Treatment responses and differences between telenutrition and the traditional consultations were made according to BMI, waist, hip and initial-waist/height ratio (iWaist), calculating for the relative risk.

Results. In 68 (29.2%) women who chose traditional attention, 9 (37.5%) dropped out, 24 (40%) failed and 35 (23.5%) were successful, showing 1.4% (1.0 SD) BMI loss, 5.8% (3.4 SD) in waist circumference, 4.5% (2.8 SD) in hip circumference and 0.04% (0.02 SD) in iWaist/height ratio. In 165 (70.8%) women who chose telenutrition, 15 (62.5%) dropped out, 36 (60%) failed and 114 (76.5%) were successful, showing 1.1% (1.0 SD) BMI loss, 5.0% (3.2 SD) in waist circumference, 3.5% (3.1 SD) in hip circumference and 0.03% (0.02 SD) in iWaist/height ratio. A significance level of p < 0.05 is considered.

Conclusion. Telenutrition has a failure or dropout risk factor of about half of the value of traditional consultation, and showed slight, statistically significant differences. This study concludes that telenutrition can support or sometimes replace traditional consultations when developing weight loss programs for obese women.

Introduction

The solutions for home care are becoming a response to the need to control the health care costs of the population. Advances in information and communication technology (ICT) have directly influenced the development of telemedicine and telecare solutions (While & Dewsbury, 2011).

In addition to introducing a change in the way medical care can be provided, telemedicine is becoming an industry that has the potential to generate billions of dollars. There is a need to create successful programs to provide clinical services that remain profitable (Kuzmar, Rizo & Cortés, 2014a).

The application of telehealth principles by registered dieticians or doctors to deliver medical nutrition therapy is termed telenutrition (Chung & Chung, 2010).

Obesity is a multifactorial disorder related to genetic background, environmental and behavioral factors, underlying diseases and socioeconomic status (Shea et al., 2012). It is known that the prevalence of obesity has experienced an alarming worldwide increase (Kuzmar, Rizo & Cortés-Castell, 2014); therefore, we can say that we are in the midst of a global obesity epidemy (Flegal et al., 2012). The physiology of obesity is based on an imbalance between caloric intake and energy expenditure (Kuzmar, Cortés & Rizo, in press). Obesity has become a very frequent topic of medical, nursing or nutrition advice (Macleod et al., 2013).

The status of overweight and obesity has connotations relating to the patient’s body image. The appearance or perception of body image is defined as the body shape that is made by the mind plus the subjective representations of physical appearance (Alwan et al., 2011). The concept of body image varies throughout one’s life, depending on the social influences and life situations that affect the behavior (Kuzmar, Rizo & Cortés, 2014b). Dissatisfaction with physical appearance related to body weight is higher in women, than in men, and varies across ethnic groups in relation to the cultural integration degree (Fallon, Harris & Johnson, 2014). Some authors find no relationship between dissatisfaction with body appearance and Body Mass Index (BMI) in obese and overweight women (Cortese et al., 2010).

Obesity has more adverse effects in women than in men, in relation to cardiovascular risk; in turn, some health professionals are reluctant to initiate treatment of the comorbidities of obesity and metabolic syndrome in women because they perceive less risk than that seen in men (Mozaffarian et al., 2011).

There are several studies which show that an overweight and obesity treatment to be performed for a few months attracts more women than men, but 42% leave for various reasons, including a lack of motivation with the results of weight loss and the economic costs (Carrasco et al., 2008). It has been demonstrated that social class, level of education, marital status, and alcohol and tobacco consumption are not regarded as influential factors in the successful outcome of treatment in overweight and obese patients (Kuzmar, Rizo & Cortés, 2014b).

It is necessary to provide all nutrition consultations with a global approach that delineates the necessary changes in eating habits, exercise and other respects in accordance with the acquired body image, health, etc., in order to produce an adherence to the acquired new habits. Socioeconomic and demographic changes are occurring very rapidly in some areas of the world and are accompanied by changes in lifestyle, dietary patterns and the epidemiological profile of prevalent diseases (Aballay et al., 2013).

A telehealth network can serve as a model for integrating health services (Dimmick et al., 2003) including telenutrition (Kuzmar, Rizo & Cortés, 2014a). Commercial industries have outpaced traditional healthcare consultation in terms of traditional approaches to weight control for electronic online delivery. Little is known about the effectiveness of telenutrition; this gap represents a barrier to developing successful, patient-based e-health applications for effective behavior change. The main objective of this paper is to determine the effectiveness of telenutrition versus a traditional healthcare consultation in a weight loss program.

Material and Methods

Subjects

A clinical intervention and e-health study was conducted among 233 (according to the World Health Organization (WHO) classification (World Health Organization, 2013)) overweight and obese women who consulted a nutrition clinic in Barranquilla (Colombia) for the purpose of nutritional assessment by telenutrition and traditional in-person consultation. They were subjected to a personalized weekly follow-up consultation over the course of 16 weeks in which food consumption patterns, and measures were registered. The inclusion criteria were female gender, voluntary assistance, the capability of internet communication; and did not exclude those with chronic diseases such as kidney failure, cancers, hypertension, diabetes and dyslipidemia, and so on that require medical follow up. This study also considered patients who tried to lose weight in the previous month or at an earlier date. Exclusion criteria were male gender and those who not qualify for inclusion criteria. In turn, alcohol or tobacco consumption did not affect actual results. The study was conducted according to Helsinki Rules pertaining to all patients informed consent.

Methods

Weight loss results in patients could be seen in 16 weeks (Kuzmar, Rizo & Cortés-Castell, 2014). Women were asked to choose which treatment they would follow: telenutrition or traditional consultation. In both groups (telenutrition and traditional consultation), the study included an initial in-person consultation with a complete medical record (date accessed, date of birth, personal identification data including email and messenger service, socioeconomic status, level of education, personal medical history, toxic precedents, etc.) and, depending on the group, a virtual or physical weekly WHO-recommended nutritional assessment (World Health Organization, 1995) (age, height and weight, waist and hip perimeter) (World Health Organization, 1995). After obtaining eating habits questionnaire (Dana-Farber Cancer Institute) responses, we made the weekly WHO-based (World Health Organization, 2014) low calorie diets. Patients who chose telenutrition were instructed and taught in their first in-person consultation how to measure and read waist, hip, and weight scales. Nutritional assessment and diet information were sent by email, and the patient controls by online internet messenger. In week sixteen, telenutrition patients were also evaluated in a physical consultation to control the results. Telenutrition patients were monitored weekly and supervised by email, messenger and computer chat, and those who chose traditional consultation were supervised by in-person weekly nutritional assessments at the clinic.

The failure criteria used was that the patient did not lose weight or did not lose measures; the success criteria was that the patient lost weight or lost measures after 16 continuous weeks of monitoring, either by telenutrition or traditional consultation.

After obtaining data, we calculated the initial and final BMI according to WHO criteria (World Health Organization, 2013) as well as weight, waist, hip, initial and final waist/height ratio loss percentages. The data were treated using IBM SPSS Statistics version 22.0 software, and checked for the relative risk. A significance level of p < 0.05 was considered using U Mann–Whitney. This study was approved by the institution SEMI-Servicios Médicos Integrados of Barranquilla, Colombia.

Results

A total of 233 women were interviewed, of whom 68 (29.2%) chose traditional consultation and 165 (70.8%) chose telenutrition. Twenty-four (15 (62.5%) telenutrition patients versus 9 (37.5%) traditional patients) dropped out the study with no known reason, representing 89.7% (209) follow-up tracking; we assumed that these patients discontinued the treatment because of lack of motivation.

Our focus groups included 233 individuals, with 165 (70.8%) in the telenutrition group and 68 (29.2%) in the traditional healthcare consultation group (Fig. 1).

Figure 1 Overweight and obesity treatment success through traditional consultation and telenutrition.

The dropout or failure risk of telenutrition versus traditional consultation is = 0.45 (95% CI [0.27–0.85]; p < 0.05).

Thirty-six (60.0%) patients who followed telenutrition versus 24 (40.0%) patients who followed the traditional consultation did not drop weight; they were considered to have failed; 114 (76.5%) patients who followed telenutrition versus 35 (23.5%) patients who followed traditional consultation had lost weight (Fig. 1). Significant differences were found in relation to comparative results (Table 2).

Table 1 gives descriptive information. As previously indicated, the objective of the study was to evaluate the effectiveness of telenutrition versus traditional nutritional consultations for obese patients measuring their weight, waist and hip loss results.

Table 1 Initial values of BMI, waist, hip and waist/height index among patients who achieved success in the traditional consultation versus those who chose telenutrition.

	Traditionalconsultation	Telenutrition	P (U Mann–Whitney)	
Initial BMI (SD)	27.3(4.0)	27.1(4.2)	0.858	
Final BMI (SD)	25.9(3.7)	26.0(3.9)	0.938	
Paired T test iBMI vs. fBMI(p)	0.000	0.000		
Initial waist (cm) (SD)	86.2(10.4)	87.7(10.3)	0.417	
Final waist (cm) (SD)	80.4(9.3)	79.7(9.4)	0.727	
Paired T test iWaist vs. fWaist (p)	0.000	0.000		
Initial hip (cm) (SD)	104.5(11.4)	105.0(8.6)	0.907	
Final hip (cm)	99.9(10.5)	101.4(8.3)	0.533	
Paired T test iHip vs. fHip (p)	0.000	0.000		
iW/Initial Height index (SD)	0.55(0.06)	0.53(0.06)	0.104	
iW/Final Height index (SD)	0.51(0.06)	0.50(0.06)	0.193	
Paired T test iW/iH vs. iW/fH(p)	0.000	0.000		

Responses from our focus group participants suggest that most patients chose telenutrition for their overweight treatment. Telenutrition gave home-based support to some participants using information and communications technology; the results are quite similar.

Table 2 shows BMI, hip, waist, and iWaist/height loss percentage in the 149 patients who have had success with traditional treatment showing 1.4%(1.0 SD) BMI loss, 5.8% (3.4 SD) in waist circumference, 4.5% (2.8 SD) in hip circumference and 0.04% (0.02 SD) in iWaist/height ratio; and in women who chose telenutrition showing 1.1% (1.0 SD) BMI loss, 5.0% (3.2 SD) in waist circumference, 3.5% (3.1 SD) in hip circumference and 0.03% (0.02 SD) in iWaist/height ratio. There is a slightly BMI% (p = 0.046) and Hip% (p = 0.040) difference in the traditional consultation versus telenutrition. No statistical differences were found in waist% (p = 0.177) and iWaist/height% (p = 0.131).

Table 2 BMI, waist, hip and initial waist/height percentage loss in patients who have achieved success in traditional consultation and telenutrition.

	Traditional consultation	Telenutrition	P (U Mann–Whitney)	
BMI Dif %(SD)	1.4(1.0)	1.1(1.0)	0.046	
Waist Dif %(SD)	5.8(3.4)	5.0(3.2)	0.177	
Hip Dif %(SD)	4.5(2.8)	3.5(3.1)	0.040	
iWaist/Height Dif %(SD)	0.04(0.02)	0.03(0.02)	0.131	

Discussion

The prevalence of obesity is especially concerning because of its association with increased risk for chronic health conditions; risk which can be mitigated with weight loss (Diabetes Prevention Program Research Group, 2002). Weight-loss programs include education and support for improving dietary self-monitoring and behavior change skills in patients increasing social support (Burke, Wang & Sevick, 2011).

Alternative overweight and obesity treatments are very popular, but although they are widely used they have not been shown to be safe and effective (Allison et al., 2001). Successful weight loss rates are highly variable in the literature (Hill & Williams, 1998) and depend on many factors (Paxton et al., 1991). No scientific data of the studied population have been found that can be used to compare to this research, making this study a pioneer in the region.

The present study used a comparative non-invasive clinical approach to determine if telenutrition can be more effective than the traditional healthcare consultation as determinants in the success of a treatment for obesity in a Caribbean population. Patients who completed treatment and lost weight by week 16 are 63.9% of the total that began the treatment, despite the great difficulties that arise in the treatment of overweight and obesity. The main limitation is that telenutrition was not allocated randomly, and because of the data this study did not use multivariate statistical analysis. The dropout rate was small (10.3%), perhaps because the initial attendance is voluntary, for aesthetic reasons without a clinical condition, or due to lack of motivation.

It will be interesting to know if these results are similar in other world populations in regard to the limitations; the authors are planning future research to complete this study.

Conclusion

Despite the fact that telenutrition was not allocated randomly and that multivariate statistical analysis was not used, patients who chose telenutrition had a failure or dropout risk factor of about half of the value of traditional consultation, with some slight statistically significant differences. Based on these results, ICT-based telenutrition can support or sometimes replace the traditional consultation when developing weight loss programs for obese women.

Additional Information and Declarations

Competing Interests

Author Contributions

Human Ethics

Data Deposition

The authors declare there are no competing interests.

Isaac E. Kuzmar conceived and designed the experiments, performed the experiments, wrote the paper, prepared figures and/or tables, reviewed drafts of the paper.

Ernesto Cortés-Castell analyzed the data, contributed reagents/materials/analysis tools, prepared figures and/or tables, reviewed drafts of the paper.

Mercedes Rizo contributed reagents/materials/analysis tools, reviewed drafts of the paper.

The following information was supplied relating to ethical approvals (i.e., approving body and any reference numbers):

1. SEMI—Servicios Médicos Integrados of Barranquilla, Colombia

2. Approval letter

The following information was supplied regarding the deposition of related data:

Safe Creative:

http://www.safecreative.org/work/1411052482611-telenutrition-vs-traditional-women.

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
