# Peer review of "Effectiveness of telenutrition in a women’s weight loss program"

_PeerJ, doi:10.7717/peerj.748_

## Round 0.1 · original submission · Major Revisions

Please address all issues raised by reviewers and provide a point-by-point response as to how each issue was addressed in the revised manuscript. Please pay particular attention to revising the Methods to ensure that the inclusion and exclusion criteria, allocation assignment and statistical methods are clear and comprehensive. Further, please ensure Results are presented in conventional format with corresponding p-values. In addition, please ensure there is discussion on how this work builds on prior studies in the Discussion section (being sure to cite previous papers and results) and provide opinions regarding on how the results of this work advances the field forward.

Reviewer 1 ·

Basic reporting

No comments.

Experimental design

See general comments.

Validity of the findings

See general comments.

Additional comments

The manuscript of Kuzmar et al. deals with an important topic. I think it is quite interesting but some issues have to be addressed:
1) Did you exclude patients with hypertension, diabetes and dyslipidemia? These chronic diseases are very prevalent in women with overweight and obesity.
2) Page 5, lines 82-83: “nutritional assessment” is repeated.
3) Page 5, lines 84-86: “Patients who followed telenutrition”. Which patients? Did you allocate some patients to telenutrition? Then, it is written that “All patients were controlled and supervised by telenutrition (email, messenger and computer chat) and in-person weekly nutritional assessment.” I am a little bit confused. What were the criteria to follow-up the patient through telenutrition?
4) Page 5, line 88: please write “the” instead of “de”.
5) Page 5, line 95: please write “Mann-Whitney” instead of “Maan-Whitney”.
6) Please state the exact values of the non significant p-values.
7) Please write “Strenghts and limitations of this study”. The main limitation would be that telenutrition was not allocated randomly. Another one, not to use multivariate statistical analysis.
8) Do the authors have any future research related to this study?
9) Please include the main limitation in the conclusion.
10) Please, state in methods that you calculate the relative risk (instead of OR).
11) 95% CI is more used than CI95%.
12) Please, indicate the exact p-value, instead of p<0.05.
13) Indicate in the objective of the abstract: female patients.
14) Please review the abstract in accordance with the rest of the comments.

·

Basic reporting

No comments

Experimental design

Methods. The authors should refer the used criteria when is considered to have been a failure and when a successful treatment.

Validity of the findings

Results. Lines 115-117 should be removed: " In order to avoid interference between the studied variables, the multinomial regression analysis was performed in failure versus success regarding telenutrition and traditional healthcare consultation", since these results are not shown.
Discussion. Results must be compared with similar studies; in case of absence of data this scientific fact must be highlighted.

Additional comments

This is a study that results of interest to address a new and very actual healthcare model.

---

## Round 0.2 · accepted · Accept

The reviewers have stated that the revisions made by the authors are acceptable.

Reviewer 1 ·

Basic reporting

No comments.

Experimental design

No comments.

Validity of the findings

No comments.

Additional comments

All the comments have been assessed correctly.

·

Basic reporting

No comments

Experimental design

No Comments

Validity of the findings

No Comments

Additional comments

No Comments